# Reconstructing the Biogeographic History of the Genus *Aurelia* Lamarck, 1816 (Cnidaria, Scyphozoa), and Reassessing the Nonindigenous Status of *A. solida* and *A. coerulea* in the Mediterranean Sea

**Alfredo Fernández-Alías \*, Concepción Marcos and Angel Pérez-Ruzafa** 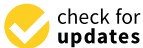

Department of Ecology and Hydrology, and Regional Campus of International Excellence "Mare Nostrum", University of Murcia, 30100 Murcia, Spain; cmarcos@um.es (C.M.); angelpr@um.es (A.P.-R.)
* Correspondence: alfredo.fernandez@um.es

**Abstract:** The genus *Aurelia* is one of the most extensively studied within the class Scyphozoa. However, much of the research was historically attributed to the species *Aurelia aurita* (Linnaeus, 1758) before the recognition of its taxonomic complexity. Initially considered cosmopolitan and globally distributed, recent phylogenetic analysis has challenged this assumption. Consequently, the current distribution of species within the genus *Aurelia* and the processes that led to this distribution remain largely unexplored. After genetically confirming that the species traditionally present in the Mar Menor coastal lagoon in the southwestern Mediterranean corresponds to *A. solida*, we compiled data on the locations where moon jellyfish species have been genetically identified and mapped these coordinates to the geological period when the genus *Aurelia* diverged from other scyphozoan genera. We propose two hypotheses to explain the disjunct distribution of certain species. The first one assumes recent human-mediated introductions, while the second posits an absence of introductions. Both hypotheses, supported by fossil and historical records, suggest a Paleo-Tethys origin of the genus *Aurelia*. Migration from this area explains most of the genus's current distribution without human intervention, being the Mediterranean Sea, where *A. solida* should be considered autochthonous, part of their natural distribution range.

**Keywords:** *Aurelia*; biogeography; nonindigenous species; paleogeography; Tethys Ocean

## 1. Introduction

The history of the genus *Aurelia* Lamarck, 1816, is intricate [1]. Although it is the most extensively studied among the scyphozoan genera [2], understanding the present distribution of its species and the mechanisms leading to their interspecific differences and segregation remains a challenge. The first original description of one of its emblematic species was based on individuals from the Baltic Sea [3] under the name of *Medusa aurita* Linnaeus, 1758. From the 18th century, two descriptions of the genus can be found: one as *Aurellia* [4] and the currently accepted as *Aurelia* [5]. In the 19th century, the number of accepted species increased to 12 [6], but Mayer [6] held the opinion that most of them were varieties of *Aurelia aurita* (Linnaeus, 1758) or *Aurelia labiata* Chamisso & Eysenhardt, 1821. Throughout the 20th century, the number of recognized species was reduced to three [7], and, until the arrival of the 21st century, despite the important differences in their responses to environmental factors found on geographically different populations, *A. aurita* has been considered a cosmopolitan and globally distributed species [6,8]. However, the genetic variability found in the phylogenetic analysis carried out by Dawson and Martin [9] indicated that *A. aurita* is part of a taxonomic complex including various species. Identifying different species based on diagnostic characters is challenging [1,10,11], but the combination of taxonomic and phylogenetic analyses has increased the number of species up to 30 in 2023 through name resurrections and new descriptions [1,11–14].

However, despite the substantial progress made in developing global biogeography databases like the Global Biodiversity Information Facility (GBIF) and the Ocean Biodiversity Information System (OBIS), their utility for cryptic species, such as those within the genus *Aurelia*, remains limited. In the case of this genus, most of the records found on GBIF and OBIS are assigned to *A. aurita*, whose previously presumed cosmopolitan distribution was reviewed in the early 21st century [9,15]. Alternatively, many records are classified under the generic label '*Aurelia* sp.' (or spp.) without species identification. Thus, there are vast geographical areas where the identification of moon jellyfishes can only be narrowed down to the genus level due to the inability to discriminate with greater precision and the scarcity of studies incorporating molecular character analyses.

Within the Mediterranean Sea, now, five species of moon jellyfish are recognized. Two of the these are classified as nonindigenous species (NIS): *Aurelia coerulea* von Lendenfeld, 1884, and *Aurelia solida* Browne, 1905. The remaining three are categorized as endemic species: *Aurelia persea* (Forsskål, 1775), *Aurelia pseudosolida* Garić & Batistić, 2022, and *Aurelia relicta* Scorrano, Aglieri, Boero, Dawson & Piraino, 2016.

*A. coerulea* was originally described from its type locality, Port Jackson in Australia. However, in the 19th century, its distribution likely extended to Japan, where it was described as *Aurelia japonica* Kishinouye, 1891. Mayer [6] considered both names as synonyms of *A. aurita*, but subsequent genetic and taxonomic analyses indicated that *A. aurita* and *A. coerulea* are different species [11,15]. The name of *A. coerulea* was given priority over *A. japonica* due to the antiquity of its description [15]. Today, *A. coerulea* is considered the most widely distributed species within the genus *Aurelia*, being found in most of the warm and temperate seas worldwide [11,15]. In the Mediterranean Sea, this species is distributed in harbors and coastal lagoons across Spain, France, and Italy, with the suggestion that this distribution is the result of ship-based introductions or aquaculture trade [11]. On the other hand, the genetic analyses of *A. coerulea*, conducted on a large spatial scale with samples collected in the Pacific, Atlantic, and Indian Oceans and the Mediterranean Sea, may lead one to think that this species could have achieved its distribution without human intervention [15]. However, these same authors clarify that oceanographic modelling, thermal tolerance of the species, and its life expectancy indicated that human-mediated transportation is necessary to explain its current distribution [15].

The type locality of *A. solida* is Maldives, but it has also been observed in the Red Sea and, within the Mediterranean basin, in the Gulf of Trieste and Porto Cesareo (Italy), Cannes (France), and Bizerte Lagoon (Tunisia) [11]. Its presence on both sides of the Suez Canal along with the type locality being in Maldives has led to the suggestion that this species is a Lessepsian migrant, although the timing and process of colonization remain unclear [11,15]. In the case of *A. solida*, the genetic analyses were conducted on a smaller spatial scale, including only individuals from the Red and Mediterranean seas [11,15], and did not examine or sequence any specimens from Maldives, the species' type locality. The resurrection of the name *A. solida* by Scorrano et al. [11] was based on the taxonomic characters mentioned in the original description by Browne [16], but these characteristics were questioned after being found in genetically distinct species [1].

*A. persea* was originally described as of *Medusa persea* Forsskål, 1775, but the description was brief and lacked details, leading to its transfer to *A. aurita* [17]. The resurrection of the name proposed by Lawley et al. [1] was based solely on the genetic sequences obtained by Mizrahi [18] from individuals collected in Haifa Bay, Israel. Unfortunately, the type material was unavailable to conduct further taxonomic or genetic analyses [1].

*A. pseudosolida* was described based on a single specimen, a ripe female, in the locality of Rovinj (Croatia) collected during a bloom, where it coexisted with *A. solida* and *Mnemiosis leidyi* A. Agassiz, 1865. The species was determined based on taxonomic and genetic characters, but this still constitutes the only record of the species [13].

*A. relicta* was described from multiple individuals collected in the Mjlet lakes (Croatia). The known distribution of the species is limited to this location, where it has been suggested to be a local endemism [11].

The current understanding of the biogeography of the genus *Aurelia* in the Mediterranean Sea, with the classification of *A. coerulea* and *A. solida* as NIS, and the distribution of *A. persea*, *A. pseudosolida*, and *A. relicta* limited to their type localities, suggests that before the opening of the Suez Canal (1869–1879), this genus, once considered globally distributed [6,17], was restricted to specific locations within the Mediterranean Sea. However, the records of the genus *Aurelia* before the opening of the Suez Canal, as documented by Péron and Lesueur [4] or Lamarck [5], were not discussed by either Dawson et al. [15] or Scorrano et al. [11].

Consequently, there is a need to reassess the NIS status of *A. coerulea* and, particularly, of *A. solida* within the Mediterranean waters. To accomplish this, we employed a multistep approach ranging from a local to a global perspective. First, we conducted genetic identification of the moon jellyfish in the Mar Menor coastal lagoon (SE Spain), a location where the individuals have always been considered indigenous [19], as *A. solida*. Secondly, we reviewed the records of the genus *Aurelia* in the Mediterranean Sea prior to and during the opening of the Suez Canal. Finally, we analyzed the current distribution of the species within the *Aurelia* genus and undertook a paleo-reconstruction of its biogeography to formulate hypotheses that explain the present distribution.

## 2. Material and Methods

### 2.1. Study Site and Genetic Identification of the Aurelia Species from the Mar Menor

The Mar Menor coastal lagoon is a transitional ecosystem, between land and sea, located on the southeast coast of Spain (Figure 1). This unique ecosystem has been under the influence of significant anthropic pressures, which have led to the development of a eutrophication process marked by frequent jellyfish blooms and, ever since 2016, regular dystrophic crises [20,21]. Due to this and the changes that the lagoon undergoes as a consequence of anthropogenic action, the Mar Menor has been the subject of monitoring of the environment and the quality of its waters through various scientific projects since 1997 [20–23]. The scyphozoan assemblage in this area consists of six different species [21], among which only the moon jellyfish, previously identified as *Aurelia aurita* [19] or *Aurelia* sp. [21,24], is considered autochthonous.

In the broader Mediterranean Sea region, species belonging to the genus *Aurelia* typically exhibit spatial segregation [11]. However, the coexistence of *A. solida* and *A. pseudosolida* in both time and space has been observed in a specific location [13]. This emphasizes the need to increase sampling of individuals within a given habitat, on both spatial and temporal scales, to assess the species residing in that particular area.

To identify the moon jellyfish from the Mar Menor coastal lagoon, we collected 24 individuals between 2020 and 2022 from different points within the ecosystem. Additionally, to investigate whether a replacement of the native species occurred within the context of a change in *Aurelia*'s blooming pattern [23], we obtained one specimen from the zoology department at the University of Murcia, collected in June 1987 (Figure 1). The date and coordinates of collection along with the accession number of the genetic sequences deposited in GenBank are provided in Table 1.

DNA extractions were carried out using the kit QIAamp DNA Mini Kit (QIAGEN, Hilden, Germany), following manufacturer's instructions. The DNA concentration was measured with a NanoDrop2000 Spectrophotometer (Thermo Scientific, Willmington, DE, USA) and adjusted to 10 ng/μL. Two different genetic markers, one mitochondrial (cytochrome *c* oxidase subunit I or COI) and one nuclear (large ribosome subunit 28S or 28S rDNA), were amplified and sequenced. In both cases, the PCR mixture consisted of 1 μL template DNA, 1× reaction buffer, 0.2 mM dNTPs, 0.5 μM primer (forward and reverse), and 0.4 U of Taq polymerase (MyTaq DNA polymerase by Bioline, Cincinnati, OH, USA). COI was amplified with the primers LCOjf [15] and HCO2198 [25] following the profile described by Piraino et al. [26]. The 28S rDNA was amplified with the primers Aa_L28S_21 and Aa_H28S_1078 [27] following the profile described by Bayha et al. [27].

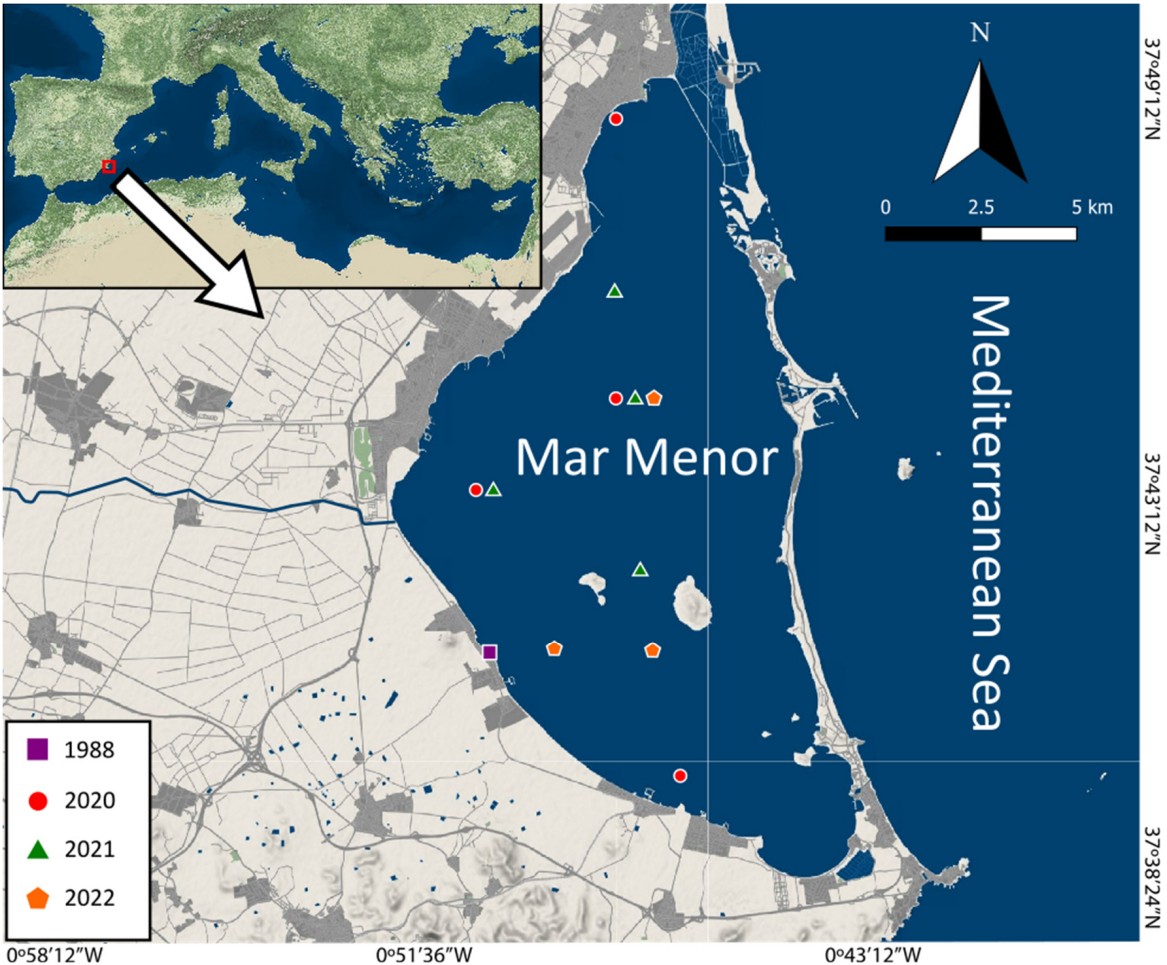

**Figure 1.** Mar Menor location and *Aurelia* sampling locations.

**Table 1.** Date and coordinates of moon jellyfish collection in the Mar Menor coastal lagoon. Accession number for the COI and 28S sequences deposited in GenBank. Empty cell denotes the absence of the sequence.

| Sample | Date | Latitude | Longitude | COI | 28S |
|---|---|---|---|---|---|
| I00 | June 1987 | 37°41.159′ N | 0°47.836′ W | | OR724094 |
| I01 | 28 April 2020 | 37°39.374′ N | 0°47.291′ W | OR727336 | OR724077 |
| I02 | 28 April 2020 | 37°39.374′ N | 0°47.291′ W | OR727337 | OR724078 |
| I03 | 28 April 2020 | 37°39.374′ N | 0°47.291′ W | | OR724079 |
| I04 | 28 April 2020 | 37°39.374′ N | 0°47.291′ W | OR727338 | OR724080 |
| I05 | 28 April 2020 | 37°39.374′ N | 0°47.291′ W | OR727339 | OR724081 |
| I06 | 28 April 2020 | 37°43.405′ N | 0°49.813′ W | OR727340 | OR724082 |
| I07 | 28 April 2020 | 37°43.405′ N | 0°49.813′ W | OR727341 | OR724083 |
| I08 | 28 April 2020 | 37°44.691′ N | 0°47.292′ W | | OR724084 |
| I37 | 15 June 2020 | 37°47.777′ N | 0°47.531′ W | OR727326 | OR724085 |
| I40 | 15 June 2020 | 37°47.777′ N | 0°47.531′ W | OR727325 | OR724086 |
| I83 | 15 June 2020 | 37°47.777′ N | 0°47.531′ W | OR727324 | OR724087 |
| I84 | 15 June 2020 | 37°47.777′ N | 0°47.531′ W | OR727323 | OR724088 |
| I88 | 4 February 2021 | 37°46.193′ N | 0°47.655′ W | OR727335 | OR724070 |
| I89 | 4 February 2021 | 37°46.193′ N | 0°47.655′ W | OR727334 | OR724071 |
| I90 | 4 February 2021 | 37°44.691′ N | 0°47.292′ W | OR727333 | OR724072 |
| I91 | 4 February 2021 | 37°44.691′ N | 0°47.292′ W | OR727332 | OR724092 |
| I92 | 4 February 2021 | 37°43.405′ N | 0°49.813′ W | OR727331 | OR724093 |

**Table 1.** *Cont.*

| Sample | Date | Latitude | Longitude | COI | 28S |
|--------|------|----------|-----------|-----|-----|
| I93 | 4 February 2021 | 37°43.405′ N | 0°49.813′ W | OR727330 | OR724073 |
| I94 | 4 February 2021 | 37°42.269′ N | 0°47.202′ W | OR727329 | OR724074 |
| I95 | 4 February 2021 | 37°42.269′ N | 0°47.202′ W | OR727328 | OR724075 |
| I96 | 4 February 2021 | 37°42.269′ N | 0°47.202′ W | OR727327 | OR724076 |
| I117 | 31 May 2022 | 37°41.141′ N | 0°47.977′ W | | OR724089 |
| I118 | 31 May 2022 | 37°41.155′ N | 0°48.730′ W | OR727322 | OR724090 |
| I119 | 31 May 2022 | 37°44.691′ N | 0°47.292′ W | OR727321 | OR724091 |

The size of the PCR products was checked in an agarose gel (0.8%), and both strains were sequenced at the University of Murcia facilities. Electropherograms were manually inspected and edited using SnapGene (www.snapgene.com, accessed on 12 July 2023). Sequences' identity was confirmed by BLASTn against the nucleotide database (GenBank) of the National Center for Biotechnology Information (NCBI, www.ncbi.nlm.nih.gov, accessed on 12 July 2023). Accession numbers to the sequences deposited in GenBank are provided on Table 1.

To determine if all the *Aurelia* individuals collected in the Mar Menor belonged to the same species, genetic distances between specimens were calculated using the Kimura-2-parameters (K2P) evolutionary model [28] and bootstrap (2000 replicates) to verify phylogenies [29]. Species determination was carried out by analyzing genetic distances using the criterion of distances greater than 6% for the COI [11] and was later confirmed by the construction of phylogenetic trees. That threshold was not exceeded for the moon jellyfishes in the Mar Menor (Table S1), and, for the subsequent analysis, they were treated as a single species.

Moreover, the sequences obtained by Scorrano et al. [11] to resolve species in the Mediterranean Sea were added to the sequences produced in this study. Two sequences of *Pelagia noctiluca* Forsskål, 1775, were used as an outgroup (accession numbers: KJ573419 for COI; KJ573408 for 28S). We conducted a multiple alignment using ClustalW for each genetic marker and trimmed the sequences to the shortest one. We established 6 groups: *Aurelia* sp. From Mar Menor, *A. aurita*, *A. coerulea*, *A. solida*, *A. relicta*, and *P. noctiluca*, and calculated the genetic distances with the K2P evolutionary model and 2000 replicates of bootstrap for both genetic markers, COI and 28S. Finally, maximum likelihood (ML), with 2000 bootstrap replicates and K2P as the evolutionary model, and Bayesian inference (BI), with evolutionary model selection under AIC in jModelTest 2.1.7 [30], phylogenetic trees were constructed. The BI phylogenetic tree was constructed in MrBayes 3.2.7 [31], the rest of the phylogenetic analyses were performed using MEGA7 [32], and software FigTree v1.4.3 was used to represent the phylogenetic tree.

### 2.2. Mediterranean Species of Aurelia Prior to and during the Construction of the Suez Canal

The construction of the Suez Canal, which took place between 1859 and 1869, complicates the systematic review of the records using standard methodologies. The earliest works retrieved from the literature databases such as SCOPUS and Web of Science (WOS) through the search algorithm 'Aurelia AND jellyfish' date back to 1949 and 1972, respectively. This prevents the use of the methodology PRISMA (preferred reporting items for systematic reviews and meta-analyses) [33] to collect information contemporaneous with the construction of the Suez Canal. We thus conducted a selection and analysis of the literature from the Biodiversity Heritage Library (www.biodiversitylibrary.org, accessed on 25 October 2023) in search records related to *Aurelia* prior to the opening of the Suez Canal (Table 2).

**Table 2.** The Biodiversity Heritage Library literature selection for the determination of Mediterranean biogeography of the *Aurelia* genus prior to and contemporaneous with the construction of the Suez Canal.

| References (Chronological Order) |
| --- |
| Linnaeus, C. *Systema Naturae per tria Naturæ, Secundum Classes, Ordines, Genera, Species, Cum Characteribus, Differentiis, Synonymis, Locis*, 10th ed.; Impensis Direct Laurentii Salvii: Stockholm, Sweden, **1758**; Volume 1, p. 824. [3] |
| Forsskål, P. *Descriptiones Animalium, Avium, Amphibiorum;* Mölleri: Copenhagen, Denmark, **1775**. [34] |
| Gmelin, J.F. *Caroli a Linne. Systema Naturae Per Regna Tria Naturae, Secundum Classes, Ordines, Genera, Speciescum Characteribus, Differentiis, Synonymis, Locis*; Delamolliere, J.B., Ed.; Lyon, France, **1789**. [35] |
| Péron, F.; Lesueur, C.A. Tableau des caractères génériques et spécifiques de toutes les espèces de méduses connues jusqu'à ce jour. *Ann. Du Muséum Natl. D'histoire Nat. De Paris* **1810**, *14*, 325–366. [4] |
| Lamarck, J.B.M. *Histoire Naturelle des Animaux Sans Vertèbres*; Paris, France, **1816**. Volume 2, p. 568. [5] |
| Chamisso, A.; Eysenhardt, C.G. De animalibus quibusdam e classe vermium Linneana, in circumnavigatione Terrae, auspicante Comite N. Romanoff, duce Ottone di Kotzebue, annis 1815–1818 peracta, observatis Fasciculus secundus, reliquos vermes continens. *Nova Acta physico-medica Academiae Cesareae Leopoldino-Carolinae 10,* **1821**. [36] |
| Lesson, R.P. Zoologie. In *Voyage Autour du Monde: Exécuté Par Ordre du roi, sur la Corvette de Sa Majesté, la Coquille, Pendant les Années 1822, 1823, 1824, et 1825*; Duperrey, M.L.I. Paris, France, **1830**. [37] |
| Brandt, J.F. Prodromus descriptionis animalium ab H. Mertensio observatorum: Fascic. I. Polypos, Acalephas Discophoras et Siphonophoras, nec non Echinodermata continens. In *Recueil des Actes de la Séance Publique de l'Académie Impériale des Sciences de St. Pétersbourg*; Académie Impériale des Sciences: Saint Petersburg, Russia, **1835**. [38] |
| Agassiz, L. *Contributions to the Natural History of the United States of America*; Little, Brown and Company: Boston, MA, USA, **1862**. [17] |
| Haeckel, E. System der Ascrapeden. *Monographie der Medusen*; Fisher, J.G., Ed.; Germany, **1879**. [39] |
| von Lendenfeld, R. The scyphomedusae of the southern hemisphere. Part I-III. *Proc. Linn. Soc. New South Wales* **1884**, *9*, 259–306. [40] |
| Vanhöffen, E. Untersuchungen über Semäostome und Rhizostome Medusen; *Bibliotheca Zoologica*, **1888**. [41] |
| Agassiz, A.; Mayer, A.G. Acalephs from the Fiji Islands. *Bull. Mus. Comp. Zool., 32*, **1899**. [42] |
| Bigelow, H.B. Medusae from the Maldive Islands. *Bull. Mus. Comp. Zool, 39*. **1904**. [43] |
| Browne, E.T. Scyphomedusae. In *The Fauna and Geography of the Maldive and Laccadive Archipelagoes*; Gardiner, I.S., Ed.; University Press: Cambridge, UK, **1905**; Volume 2 (Suppl. 1) [16] |
| Mayer, A.G. *Medusae of the World. Vol III, The Scyphomedusae*; Carnegie Institution of Washington: Washington, DC, USA, **1910**. [6] |

### 2.3. Present Biogeography of the Genus Aurelia

Given the recent resurrection and new species description within the genus *Aurelia* [1,11–14] and the absence of taxonomic diagnosis characters [1], the biogeography of the species should be restricted to the locations where genetic determinations were made. In this section, we implemented a modified version of the PRISMA method [33] to systematically review the locations with available genetic data on *Aurelia*.

First, current accepted species [44] and the not-yet-described *sensu* Lawley et al. [1] and *sensu* Moura et al. [14] were individually searched on the nucleotide databases GenBank and Barcode of Life Data System (BOLD, https://boldsystems.org/, accessed on 20 July 2023). For each sequence, we retrieved specimen collection coordinates. *A. aurita* was excluded from this step given the high number of nonreviewed sequences deposited before the determination of the cryptic nature of the species. Next, we extracted the coordinates from the sequences obtained on global-scale genetic analyses of the genus *Aurelia* [1,14] and removed the duplicates.

Finally, we individually searched for the name of each accepted species on SCOPUS, selecting relevant works and extracting the coordinates based on the inclusion criteria of genetic confirmation of the species. Records from species whose belonging could not be ascertained, referred to as *confer* (cf.) (compare with), were discarded when they did not contribute to the distribution of the species they were compared against. Among

the currently accepted species within the genus *Aurelia* [44], *Aurelia colpota* Brandt, 1835, *Aurelia maldivensis* Bigelow, 1904, and *Aurelia vitiana* Agassiz & Mayer, 1899, lack any genetic records and were, therefore, not included in the analysis. The list of coordinates for each species can be found in Table S2.

Phylogenetic analyses on a global scale revealed the presence of different evolutive lineages of *Aurelia* [1,14]. In accordance with these analyses, we have defined a color pattern for the representation of the lineages in four groups (Figure 2), which will be maintained in the representation of both present and past biogeography. The lineages were named according to their current distribution (see Section 3.3) as 'Boreal', 'Atlanto-Mediterranean', 'Indo-Pacific', and 'Western Atlantic'.

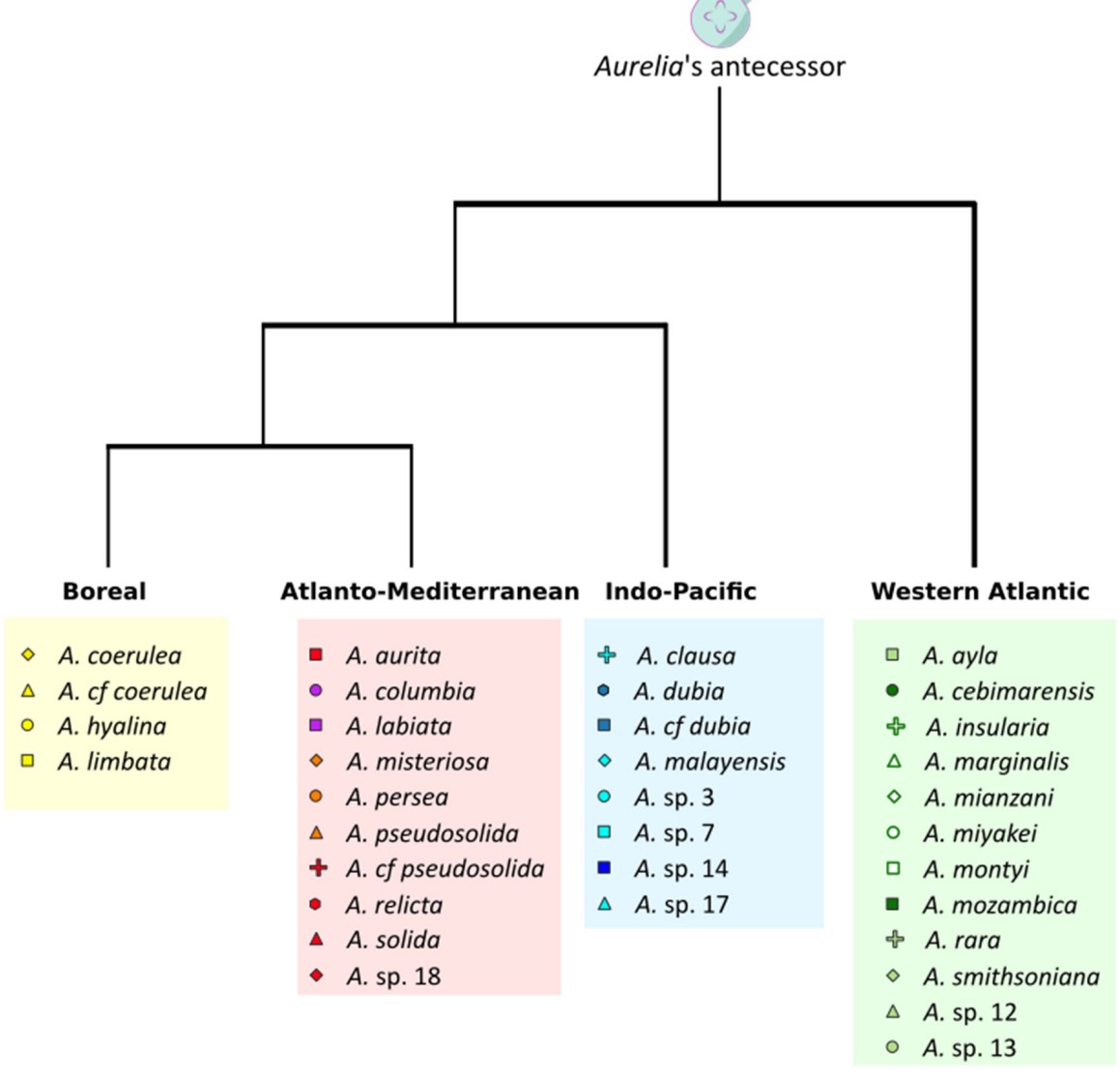

**Figure 2.** Cladogram of the *Aurelia* genus based on the concatenated genetic analysis (COI + 28S + ITS5.8) of Lawley et al. [1], complemented with Moura et al. [14].

*2.4. Paleogeography of the Genus Aurelia*

The reconstruction of the paleogeographic distribution of the genus *Aurelia* presents several challenges due to the limited fossil record [45] and the potential impact of human activities on its current distribution [15]. To address these challenges, it is necessary to establish certain conditions for the formulation of the hypotheses.

The genus *Aurelia* diverged from other scyphozoan lineages approximately 300–400 million years ago (Mya), and lineage separations occurred 250–200 Mya [45]. This suggests that the *Aurelia* genus emerged and started to diverge into distinct lineages during the time when the supercontinent Pangea separated the Paleo-Tethys from the Panthalassa Ocean [46,47]. As such, the origin of the *Aurelia* genus could be located in either the Paleo-Tethys or in Panthalassa. By rotating the coordinates of the current locations of different *Aurelia* lineages, we can place them in the Paleo-Tethys approximately 270 Mya ago but not in Panthalassa. Therefore, we used Paleo-Tethys as its origin for constructing our hypotheses. The coastline of the Paleo-Tethys, at the moment of its formation (300 Mya), included what is now Europe, the northern part of Africa, the Arabian Peninsula, India, Australia, and China [47].

To explain the current distribution of different species within the genus *Aurelia*, assuming a common ancestor and a Paleo-Tethys (or Tethys) origin 300 Mya, we formulated two distinct hypotheses. The first hypothesis attributes the disjunct distribution of some species and lineages exclusively to recent anthropogenic introductions. The second, in contrast, explains disjunct distributions exclusively by natural causes and paleogeographic processes.

For the reconstruction of the paleogeography of *Aurelia*, we retrieved past Earth maps from the PALEOMAP PaleoAtlas for Gplates collection dating back 270 Mya, 237 Mya, 195 Mya, 150 Mya, 94 Mya, 50 Mya, 14 Mya, and 5 Mya. The rotation of the current *Aurelia* coordinates to million years in the past was performed on the website of the Paleolocation Mapping Service (www.paleolocation.org, accessed on 15 September 2023).

### 3. Results

#### 3.1. Determination of the Aurelia Species from the Mar Menor Coastal Lagoon

The genetic distance between the individuals sampled at the Mar Menor coastal lagoon is less than 0.3% for the 28S rDNA and lower than 2.1% for COI (Table S1). Therefore, the criteria for distinct species determination, which requires a genetic distance higher than 6% for the COI [11], is not met. Consequently, all the individuals should be grouped under the name 'Mar Menor'.

Pairwise calculation of genetic distances between the groups from Scorrano et al. [11] and the 'Mar Menor' group reveal that the individuals collected in the coastal lagoon belong to the species *A. solida*, as they have genetic distances of 0.1% for 28S rDNA and $2.1 \pm 0.5\%$ in the case of COI (Table 3). This analysis matches the results of the phylogenetic trees conducted for both genes. In the phylogenetic trees, both maximum likelihood and Bayesian inference, the 'Mar Menor' individuals are clustered with *A. solida* with a bootstrap (ML) and posterior probability (BI) support of 98% in the case of the 28S rDNA and 100% for COI (Figure 3, Figures S1 and S2).

**Table 3.** Estimates of evolutionary divergence over sequence pairs between groups. The number of base substitutions per hundred sites from averaging over all sequence pairs between groups $\pm$ standard error are shown: 28S: above diagonal; COI: below diagonal.

| | Mar Menor | *A. solida* | *A. aurita* | *A. relicta* | *A. coerulea* | Outgroup |
|---|---|---|---|---|---|---|
| **Mar Menor** | | $0.1 \pm 0$ | $2.7 \pm 0.6$ | $2.4 \pm 0.5$ | $2.1 \pm 0.5$ | $19.0 \pm 1.7$ |
| *A. solida* | $2.1 \pm 0.5$ | | $2.7 \pm 0.6$ | $2.4 \pm 0.5$ | $2.1 \pm 0.5$ | $19.0 \pm 1.7$ |
| *A. aurita* | $22.3 \pm 2.3$ | $22.0 \pm 2.3$ | | $4.1 \pm 0.7$ | $3.7 \pm 0.7$ | $19.1 \pm 1.7$ |
| *A. relicta* | $22.2 \pm 2.4$ | $21.7 \pm 2.4$ | $20.5 \pm 2.2$ | | $2.9 \pm 0.5$ | $20.3 \pm 1.7$ |
| *A. coerulea* | $20.2 \pm 2.1$ | $20.2 \pm 2.1$ | $21.0 \pm 2.3$ | $21.9 \pm 2.4$ | | $19.7 \pm 1.7$ |
| **Outgroup** | $28.4 \pm 2.8$ | $28.4 \pm 2.8$ | $28.0 \pm 2.7$ | $27.5 \pm 2.8$ | $27.7 \pm 2.7$ | |

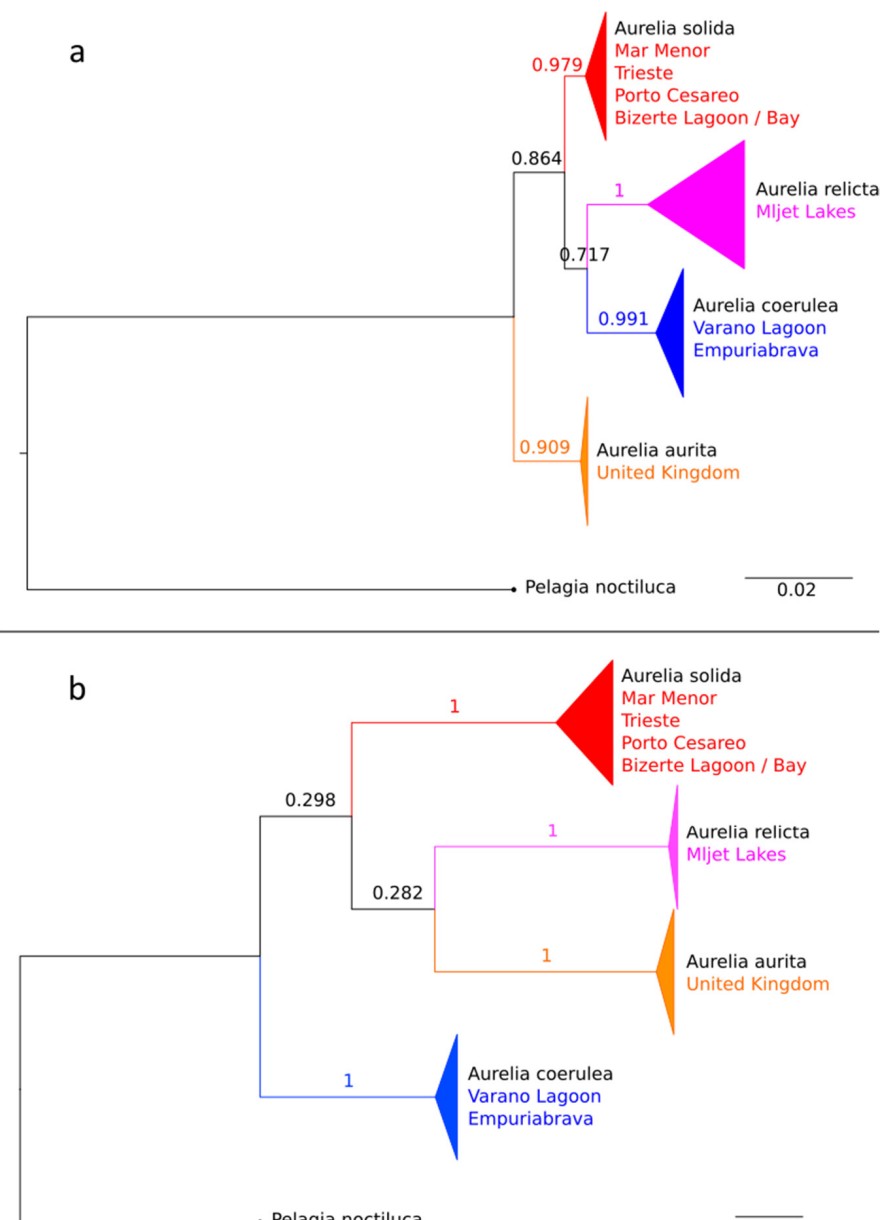

**Figure 3.** Evolutionary history, as inferred by maximum likelihood, of the genus *Aurelia* in the Mediterranean Sea. (**a**) Phylogenetic tree for the 28S rDNA. (**b**) Phylogenetic tree for the COI genetic marker. Numbers above the tree branches indicate bootstrap support for each branch. Clades were collapsed for a support higher than 0.9 at species level to ease visualization. The tree is drawn to scale, with branch length measured in the same units as the evolutionary distances used for their calculation. Non-collapsed ML and BI trees are provided in Figure S1 and Figure S2, respectively.

*3.2. Mediterranean Species of Aurelia Prior to and during the Construction of the Suez Canal*

We found several records of individuals belonging to the *Aurelia* genus in the Mediterranean Sea dating back to 1775 (Table 4), which is 84 years before the construction of the Suez Canal began.

The first reference to a specimen belonging to this genus can be traced to Linnaeus's book, *Systema Naturae*, where it was identified as *Medusa aurita*, referring to individuals from the Baltic Sea [3]. In this edition of *Systema Naturae*, the Mediterranean Sea was not listed as the habitat for the type species, *A. aurita*. However, it was later recognized as such in Gmelin's edition [35].

**Table 4.** Records of the *Aurelia* genus in the Mediterranean Sea prior to and close to the construction of the Suez Canal.

| Reference | Species (as Indicated) | Location |
|---|---|---|
| Forsskål, 1775 [34] | *Medusa persea* | Mediterranean Sea |
| | *Medusa cruciata* | Mediterranean Sea |
| Gmelin, 1789 [35] | *Medusa tyrrhena* | Thyrrenian Sea |
| | *Medusa persea* | |
| | *Medusa crucigera* | |
| Péron & Lesueur, 1810 [4] | *Aurellia phosphorica* | Strait of Messina |
| | *Aurellia amaranthea* | Naples |
| | *Aurellia rufescens* | Mediterranean Sea |
| Lamarck, 1816 [5] | *Aurelia phosphorea* | Strait of Messina |
| | *Aurelia tyrrhena* | Naples |
| | *Aurelia crucigera* | Mediterranean Sea |
| Agassiz, 1862 [17] | *Aurelia aurita* | Mediterranean Sea |
| Haeckel, 1879 [39] | *Aurelia aurita* | Atlantic coast of Europe and Mediterranean Sea |
| Mayer, 1910 [6] | *Aurelia aurita* | Atlantic coast of Europe and Mediterranean Sea |
| | *Aurelia cruciata* (a variety of *A. aurita*) | Atlantic coast of Spain and Mediterranean Sea |

Furthermore, Forsskål [34] documented several Mediterranean species that would later be transferred to the genus *Aurelia* at the moment of its description [4,5] or in Agassiz [17]. Haeckel [39] and Mayer [6] considered *A. aurita* as the sole moon jellyfish species inhabiting the Mediterranean Sea, and, previously, Agassiz [17] suggested that, in Europe, all the described species under the name of *Aurelia* comprised two varieties of *A. aurita*: one from northern Europe and another found in southern Europe and the Mediterranean Sea.

The distribution of this genus before the opening of the Suez Canal cannot be determined precisely, but it can be affirmed that it was, at least, present along the coast of Italy as mentioned by Péron and Lesueur [4] and Lamarck [5]. It is worth noting that, by that time, Agassiz [17] and Mayer [6] believed the species to be distributed throughout the entire Mediterranean coastline.

*3.3. Present Biogeography of the Genus Aurelia*

The distribution of the genetically identified lineages on a large spatial scale for the genus *Aurelia* (Figure 2) is supported by the geographic distribution of the species within the proposed lineages (Figure 4). The western Atlantic lineage, consisting of 12 species, is primarily distributed within the area encompassing the Gulf of Mexico, the east coast of South America, and the Gulf of California. There is also a disjunct distribution extending to Mozambique, where *Aurelia mozambica* Brown & Gibbons, 2021, can be found, and Thailand, where *Aurelia miyakei* Lawley, Gamero-Mora, Maronna, Chiaverano, Stampar, Hopcroft, Collins & Morandini, 2021, is present.

The Indo-Pacific lineage includes eight species and is mainly situated in the western Pacific Ocean, extending to the east as far as the Arabian Peninsula and to the west as far as Panama. The Atlanto-Mediterranean lineage is primarily located in the Mediterranean Sea and the northern Atlantic, with extensions through the Arctic Ocean to the east until the White Sea (*A. aurita*) and to the east, including Hudson Bay (*A. aurita*), the Gulf of Alaska, and the northeastern Pacific Ocean (*Aurelia columbia* Lawley, Gamero-Mora, Maronna, Chiaverano, Stampar, Hopcroft, Collins & Morandini, 2021, and *A. labiata*). *A. aurita* is also found in the northwestern Pacific Ocean and in South America, ranging from Chile to Ushuaia (Argentina). The boreal lineage is distributed across the northern hemisphere, encompassing the Arctic, Atlantic, and Pacific Oceans and Mediterranean Sea. In the southern hemisphere, the presence of this lineage is limited to Australia.

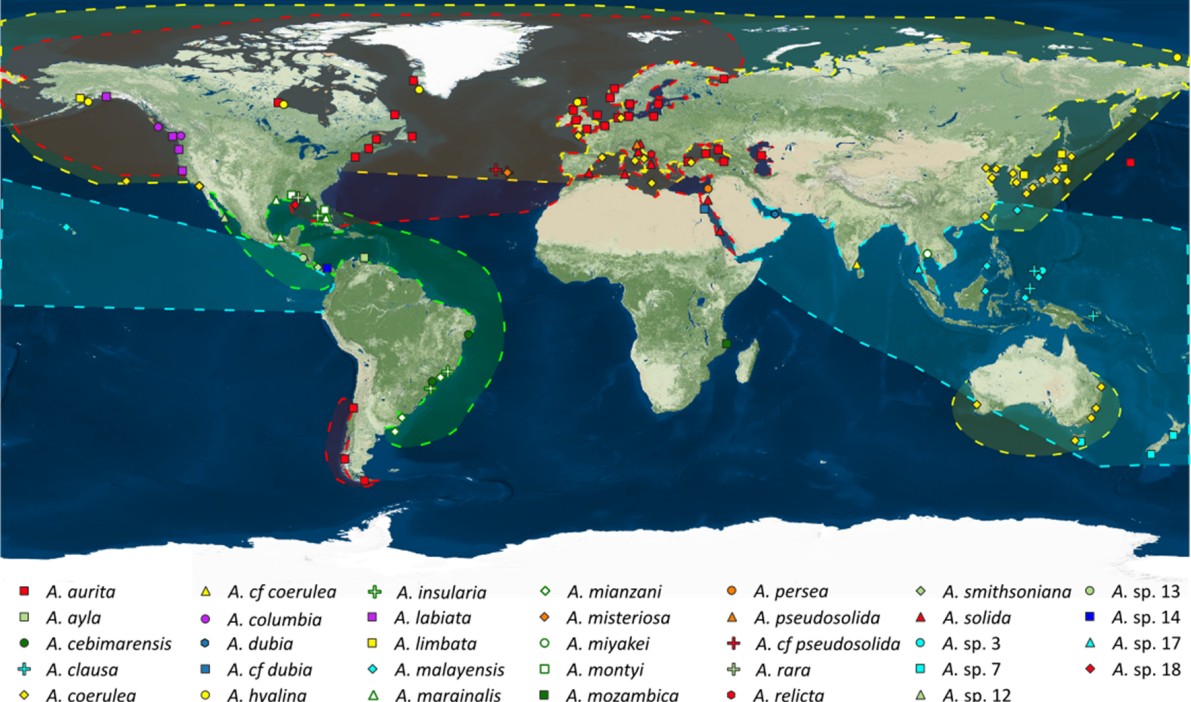

**Figure 4.** Present biogeographic distribution of the genus *Aurelia* based on genetically identified individuals. Dashed areas indicate the main distribution areas of the different lineages. Yellow: boreal; Red: Atlanto-Mediterranean; Green: western Atlantic; Blue: Indo-Pacific.

### 3.4. Paleogeography of the Genus Aurelia

3.4.1. Hypothesis A: Disjunct Distributions Explained by Anthropic Translocations

The disjunct distribution of some species has been explained by some authors by human intervention, either by direct translocation through shipping and movement of species as a consequence of aquaculture or by facilitating their introduction due to the construction of artificial communication channels between different seas and oceans. Logically, both situations are necessarily recent and would have taken place in the last 200 years.

Late Carboniferous–Permian (305 Mya–265 Mya): *Aurelia*'s antecessor was distributed along the south coast of the Paleo-Tethys Ocean (Figure 5a).

Late Permian–Lower Triassic (265 Mya–230 Mya): The formation of Cimmeria split the Paleo-Tethys, leading to the formation of the Tethys Ocean in the southern region. *Aurelia*'s antecessor was distributed across the Tethys Ocean (Figure 5b).

Lower Triassic–Upper Jurassic (230 Mya–145 Mya): Cimmeria moved northward, and the Indo-Pacific lineage began to separate from the other lineages, to which it remained connected by *Aurelia dubia* Vanhöffen, 1888 (and *Aurelia* cf. *dubia*). Pangea began to break apart, and the western Atlantic lineage started migrating westward into the newly formed Central Atlantic Ocean. The boreal and probably the Atlanto-Mediterranean lineages migrated northward through the opening between North America and Eurasia (Figure 5c,d).

Cretaceous–Cenozoic (145 Mya–present day): The western Atlantic lineage established itself in the Gulf of Mexico and colonized the Atlantic coast of South America after the formation of the South Atlantic Ocean. The spread to the Gulf of California and Panama could have taken place between the Upper Cretaceous and the Miocene (Figure 5e–g). The Indo-Pacific lineage gradually colonized the western Pacific islands that formed during this period and, eventually, reached the eastern Pacific through Hawaii (USA). The boreal lineage's migration through the Arctic Ocean allowed it to reach the North Pacific Ocean and North America (Figure 5e–h). Its introduction to Japan could be a result of this migration or an introduction. In this hypothesis, the arrival of *A. coerulea* in Australia and the arrival of *A.* cf. *coerulea* in India are attributed to the introduction of the species. The

Atlanto-Mediterranean lineage migrated alongside the boreal lineage through the Arctic Ocean, enabling *A. aurita* to colonize Hudson Bay and the White Sea, while *A. columbia* and *A. labiata* reached the Gulf of Alaska. The presence of *A. aurita* in the North Pacific Ocean can be a consequence of either this migration or a recent introduction.

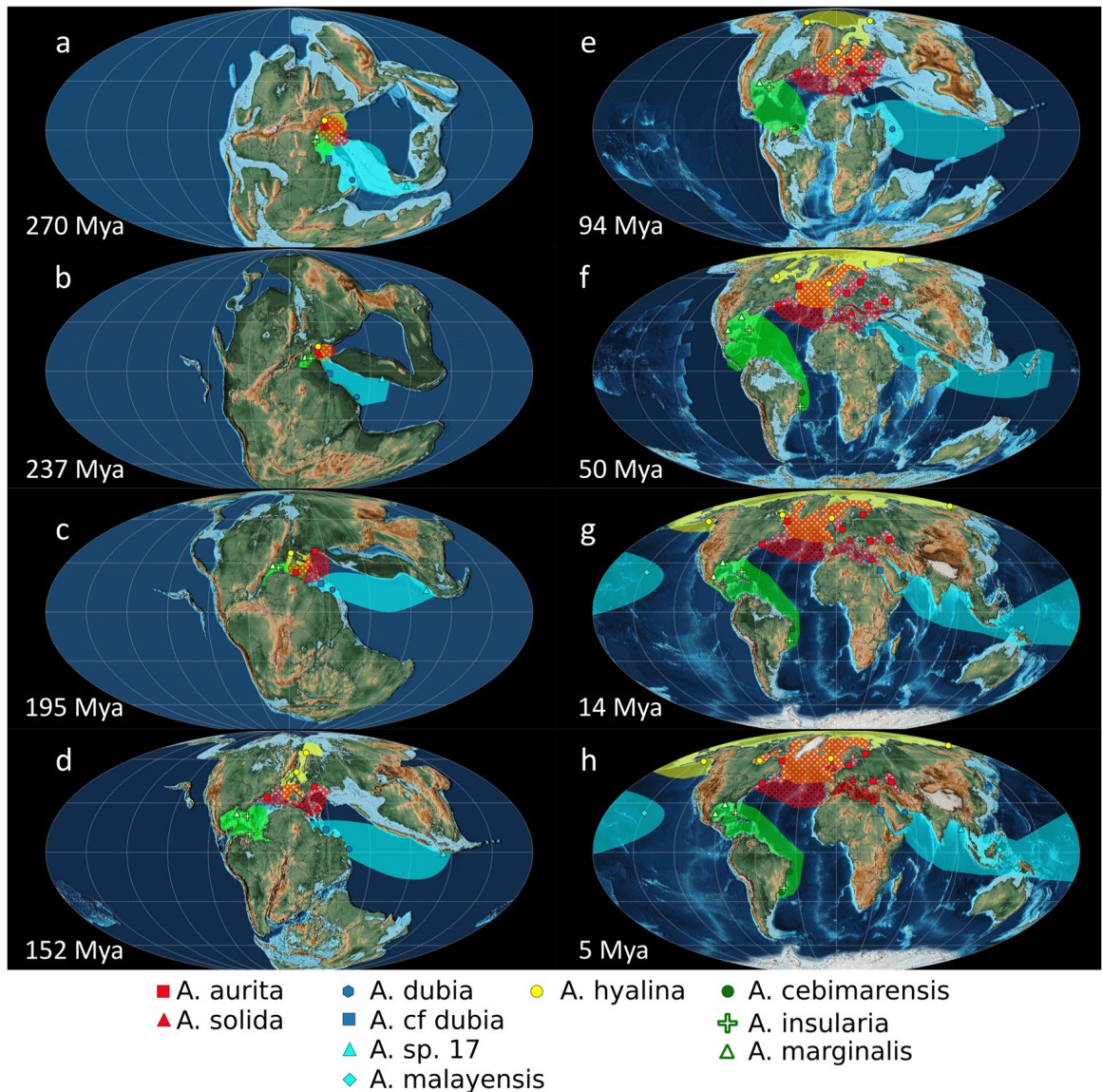

**Figure 5.** Paleogeography of the different lineages of the genus *Aurelia* explaining the present disjunct distributions through latter possible anthropic introductions.

3.4.2. Hypothesis B: Disjunct Distributions Explained by Paleogeographic Processes without Human Intervention

Late Carboniferous–Permian (305 Mya–265 Mya): *Aurelia*'s antecessor was distributed along the south coast of the Paleo-Tethys Ocean (Figure 6a).

Late Permian–Lower Triassic (265 Mya–230 Mya): The formation of Cimmeria split the Paleo-Tethys, leading to the formation of the Tethys Ocean in the southern region. *Aurelia*'s antecessor distributed across the Tethys Ocean (Figure 6b). In this hypothesis, boreal and Indo-Pacific lineages share their distribution throughout the Tethys Ocean.

Lower Triassic–Upper Jurassic (230 Mya–145 Mya): Pangea began to separate, and the western Atlantic lineage began to migrate westerly to the recently opened Central Atlantic Ocean. The boreal and, probably, the Atlanto-Mediterranean lineages migrated northward through the opening between North America and Eurasia. The Atlanto-Mediterranean

lineage migrated northward and westward (Figure 6c,d). Cimmeria moved northward and dragged the Indo-Pacific lineage along its south coast. The boreal lineage distributed over the whole Tethys Ocean (Figure 6c). The absence of present records for the boreal lineage between India and Europe (Figure 4) (imaginary line between Australia–India and South Korea–China 152 Mya), suggests a subdivision of the lineage (east–west) by local extinctions (Figure 6d). Alternatively, the boreal and the Indo-Pacific lineages could have segregated during this period, with the Indo-Pacific occupying the North Tethys Ocean and the boreal occupying the South Tethys without any subdivision of the latter.

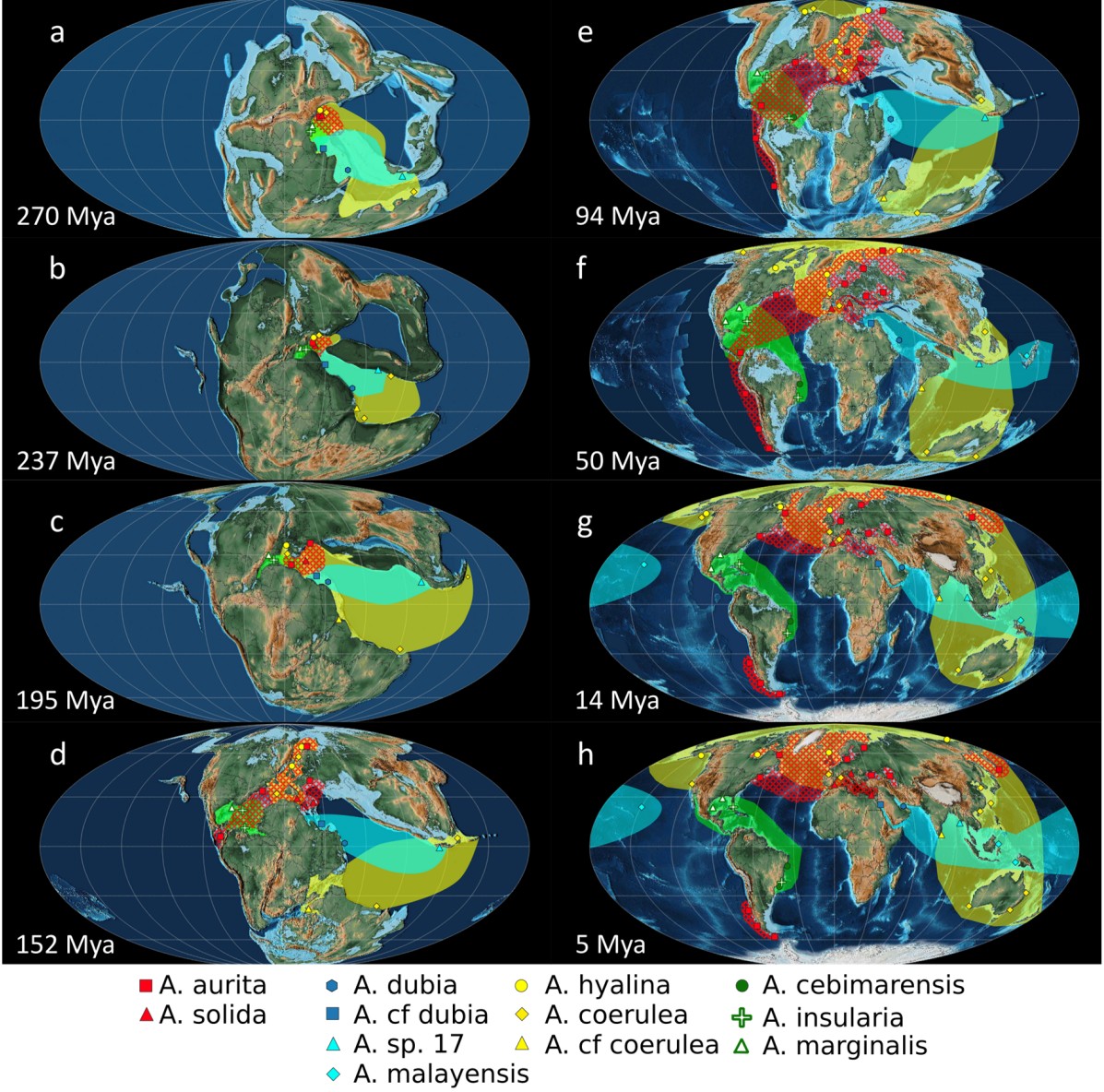

**Figure 6.** Paleogeography of the different lineages of the genus *Aurelia* explaining the present disjunct distributions without anthropic introductions.

Cretaceous–Cenozoic (145 Mya–present day): The western Atlantic lineage established in the Gulf of Mexico and colonized the Atlantic coast of South America after the formation of the South Atlantic Ocean. The spread to the Gulf of California and Panama could have taken place between the Upper Cretaceous and the Miocene (Figure 6e–g). The Indo-Pacific lineage progressively colonized the western Pacific islands, which were formed in this period, and, eventually, reached the eastern Pacific through Hawaii (USA). The boreal lineage performed a migration through the Arctic Ocean to colonize North America's Pacific

Ocean. There are three possible paths for the arrival of this lineage to the northwestern Pacific Ocean: it could have arrived via the Arctic Ocean as it did to the eastern Pacific Ocean; it could have been carried along with India when this plate separated from Australia and Antarctica to collide with Asia, or it could have already been present if its antecessor was distributed all over the Tethys Ocean (Figure 6e–g). The Atlanto-Mediterranean lineage migrated northward, through the Arctic, to reach the North Pacific Ocean and westward to reach South America's Pacific coast (Figure 6e–h). In this hypothesis, either local extinction or speciation linked perhaps to climate conditions would explain the disappearance of the Atlanto-Mediterranean lineage in the area occupied by the western Atlantic lineage.

## 4. Discussion

The reconstruction of the geographic and phylogenetic history of the genus *Aurelia* within the Mediterranean Sea (and at a global level) is a complex task due to the limited fossil record we have [45], the large number of names that were used to refer to the genus before the current denomination was established, considering the fact that the first figures and descriptions lacked detail needed to establish diagnosis characters and that no effort was made to preserve those first described specimens for further reexamination [6,17]. Even today, it is important to bear in mind that there are no clear morphologic taxonomic criteria to differentiate among moon jellyfish species and that genetic analysis is an important requirement for that task [1,10,11]. To further complicate the reconstruction of the genus's history, it should be noted that some introductions of *Aurelia* are considered to be proven and there may still be some undetected ones [1,14,15]. However, the hypotheses described here open the possibility of explaining most of the current distribution of the genus without human intervention.

According to the current interpretation, *A. aurita* is absent in the Mediterranean Sea, but it can be found in the North Atlantic Ocean and the Black Sea [15]. Within the Mediterranean Sea, the two most widely distributed species are *A. solida*, previously considered a Lessepsian migrant that colonized the Mediterranean Sea after the opening of the Suez Canal, and *A. coerulea*, whose introduction has been associated with the importation of Japanese oyster for its aquaculture in French and Italian coastal lagoons and with the maritime transport between the Pacific and Indian Oceans with the Mediterranean Sea since the second half of the 20th century [11]. Species suggested as endemic, *A. relicta*, *A. pseudosolida*, and *A. persea* [1,11,13], would be, with this interpretation, geographically restricted.

The review of the works from the 18th, 19th, and 20th centuries indicates that it was likely that this genus was distributed all over the Mediterranean Sea before the opening of the Suez Canal (Table 4). On the other hand, in the Mar Menor coastal lagoon, the autochthonous moon jellyfish previously considered as *A. aurita* [19] was identified as *A. solida* based on the genetic analysis performed on collection specimens and recently collected individuals (Figure 3; Table 3 and Table S2).

Thus, considering the different evolutionary lineages (Figure 2; [1,14]), the present biogeography of the genus based on the locations where the genetic identification of the species was conducted (Figure 4) and the autochthonous character of *A. solida* in the Mar Menor, the Indo-Pacific origin and NIS character of the Mediterranean Sea of *A. solida* are also called into question. Because of that, we have elaborated hypotheses to reconstruct the paleogeographic history of the genus *Aurelia* since the appearance of its antecessor 300 Mya [45].

### 4.1. Paleogeography of the Genus Aurelia

Despite the limited fossil record for genetic dating [45], there is some information available regarding the past distribution of the ancestors of the genus *Aurelia* [48]. The oldest fossils that might have belonged to the order Semaeostomeae, to which the genus *Aurelia* belongs, were discovered in Cambrian deposits in South China [48,49]. This region eventually became part of the eastern coast of the Paleo-Tethys around 300 Mya [47]. Determining the paleodistribution of the *Aurelia*'s antecessor, based on a fossil record that may or may not belong to the order Semaeostomeae [49] or the shifting of coordinates to

the time of Paleo-Tethys's formation, involves a high degree of uncertainty. However, this approximation becomes less speculative when we consider *Paraurelia cerinensis* Gaillard, Goy, Bernier, Bourseae, Gall, Barale, Buffetaut & Webz, 2006, found in the coastal lagoon of Cerin (France) and fossilized 150 Mya [50]. The name of *P. cerinensis* was established by Gaillard et al. [50], given the resemblance of the fossils to the impressions left by *Aurelia* in the fossilization experiments conducted by Bruton [51]. In other words, the fossil record and the rotation of coordinates to the past of the different *Aurelia* lineages support the hypothesis that the antecessor of the genus was once distributed across the Paleo-Tethys (or Tethys) Ocean.

Consequently, both suggested hypotheses, and any intermediate ones that lie between a scenario with no human intervention and one with complete human intervention to explain disjunct distribution of the lineages, start with the distribution of the ancestor of *Aurelia* across the (Paleo-)Tethys with variations on how widely certain lineages spread (Figures 5a and 6a). The presence of this genus, and its antecessors, in the region that now forms the Mediterranean Sea is reflected in its present biogeography (Figure 4), historical records (Table 4), and fossil evidence [50,52]. It is worth noting that the most recent gelatinous zooplankton fossils, which bear a strong resemblance to current *Aurelia* species [48], date back to the Eocene, 50 Mya, and were found in Verona, Italy [52]. During the Eocene, Verona was situated in a pre-Mediterranean Sea that was still connected to the Indian and Pacific Oceans in the west and to the Black Sea, Caspian Sea, and Arctic Ocean in the north [46,47].

Leaving aside *A. coerulea* and the presence of *A. aurita* on the coast of South America, both hypotheses share certain features beyond the initial distribution. Phylogenetic analyses of Lawley et al. [1] and Moura et al. [14] indicate that the Indo-Pacific and western Atlantic lineages separated early on from the clade that includes the boreal and Atlanto-Mediterranean lineages. In accordance with the two hypotheses and the period of lineage segregation calculated by Khalturin et al. [45], the lineages diverged from each other between the Triassic and the Jurassic (230–145 Mya), coinciding with the breakup of Pangea [47]. The piecewise separation of Pangea would have allowed the western Atlantic lineage to migrate westward until it reached the Gulf of Mexico and the Indo-Pacific lineage to move eastward (Figures 5 and 6). Both scenarios suggest that the boreal lineage migrated through the Arctic Ocean, supported by the current distribution of *Aurelia hyalina* Brand, 1835, around the Arctic polar circle, along with the Atlanto-Mediterranean lineage, as evidenced by the presence of *A. aurita* in Hudson Bay and the White Sea (Figure 4). In contrast, both hypotheses indicate that *A. solida* is an autochthonous Mediterranean Sea species, consistent with the phylogenetic analyses in which this species appear associated with the other endemic Mediterranean species, including *A. relicta*, *A. persea*, and *A. pseudosolida* [1,13,14].

The main differences between both hypotheses relate to the arrival of *A. aurita* in Chile and the current distribution of *A. coerulea*. The anthropic hypothesis, which accounts for greater support in the scientific community (e.g., [14,15]), suggests that *A. aurita* is an introduced species along the Pacific coast of South America. In contrast, the natural distribution hypothesis proposes that this species migrated along with the western Atlantic lineage during the breakup of Pangea and, at some point between the Cretaceous and the Miocene, it reached South America (Figure 6). In this context, it is essential to consider that in the phylogenetic analysis by Moura et al. [14], the western Atlantic lineage derives from *A. aurita*, suggesting that the disappearance of *A. aurita* from the Gulf of Mexico, as proposed by the natural distribution hypothesis, could be a climatic speciation that confined *A. aurita* to the cold waters of the South Pacific and North Atlantic Oceans (Figure 4).

*A. coerulea* presents a unique case from both the genetic and biogeographic perspectives. The type locality of the species, as indicated in the original description [40], is Port Jackson, Australia, but it was almost simultaneously described as *A. japonica* in Japan [53]. In the phylogenetic analysis performed on the genus by Dawson et al. [15], this species was referred to as *Aurelia* sp. 1, and it appeared associated with *Aurelia* sp. 10, later identified

as *A. hyalina* [1], and *Aurelia limbata* Brandt, 1835. Considering Kamchatka as the type locality of *A. limbata*, Dawson et al. [15] suggested that the presence of *Aurelia* sp. 1, later identified as *A. coerulea* by Scorrano et al. [11], in Australia, France, and California (USA) was explained by recent human introductions. It is somehow surprising that those introductions, particularly the one in Australia since it should have occurred before the species description, have not been questioned since Scorrano et al. [11] resurrected the name *A. coerulea*.

The type locality for *A. hyalina* is the Aleutian Islands (USA) and its distribution includes Hudson Bay, Greenland, the Chukotka Sea (Russia), and the north of Scotland. *A. limbata* is found between Japan and Alaska (Figure 4). Thus, both species belonging to the boreal lineage, along with *A. coerulea*, are distributed at the northernmost latitudes of our planet. The two hypotheses presented in this work suggest that the boreal lineage migrated through the Arctic Ocean, explaining its current distribution without anthropic introductions. The presence of this lineage in the Atlanto-European region and the Mediterranean Sea would be explained by the presence of its ancestor in the Tethys Ocean [50,52]. The arrival of *A. coerulea* in Japan may be a result of the Arctic migration (Figure 5), its original distribution range (Figure 6; [49]), or to the arrival of individuals from India and Australia to South Asia when India moved northward and collided with Asia [54]. The presence of *A. coerulea* in Australia may be due to the original distribution range of its ancestor (Figure 6) or could be result of a recent human introduction (Figure 5; [11,15]).

### 4.2. Limitations of the Hypotheses

Both of the hypotheses we have constructed, as well as any intermediate ones, are subject to a high degree of uncertainty and require further research. This will involve investigating the overlap between the current thermal tolerance of the species and past climate conditions, conducting in-depth population structure analysis, or analyzing the role that coastal lagoons and semienclosed coastal seas may have played in the speciation and diversification of the genus *Aurelia*, which, in fact, is characteristic of these environments. The restricted connectivity of populations and the environmental variability of coastal lagoons, intensified by past climate events such as the Messinian crises in the Mediterranean, could explain the high number of species in relatively small areas, such as the Gulf of Mexico or the Mediterranean. These efforts go beyond the objectives of this study, but our outcomes enlarge the spectrum of unexplored research avenues.

None of the approximations satisfactorily explain the presence of two species from the western Atlantic lineage in the Indian Ocean (*A. mozambica*) and the western Pacific (*A. miyakei*). Both are currently restricted to their respective type localities: Mozambique for *A. mozambica* and Thailand in the case of *A. miyakei* (Figure 4). The genetic sequences from Africa, excluding that of *A. mozambica*, are limited to the Mediterranean and Red Seas. However, it is worth noting that Africa is also the continent where our knowledge of gelatinous zooplankton is more limited [55].

The African records of the genus *Aurelia* include Sintra Bay (Western Sahara) [56], Mozambique [12], the coastline of the Gulf of Guinea [55], and, from the biogeographic databases GBIF and OBIS, we can assume the Canary Islands (Spain) and Madagascar. On the other side, the absence of the genus has been reported in the coastal waters close to the upwelling of the Benguela current [55,57]. The non-Mediterranean African records appear as *A. aurita*, *A. solida*, and *A. colpota*, but the genetic analysis indicates that the moon jellyfishes from Africa do not belong to any of the already sequenced species but to a currently undescribed species [55]. The addition of new species may modify our interpretation, but it is also possible that the current western Atlantic lineage's original distribution was similar to the boreal and the Indo-Pacific ones, but it disappeared from the Mediterranean Sea under unfavorable conditions, which could have been caused by the Messinian salinity crisis.

The Indo-Pacific lineage lacks any genetic record between the Andaman Sea, Thailand (*Aurelia* sp. 17), and the western Indian Ocean (*A. dubia* and *A.* cf. *dubia*). This fact can be

associated with a scarcity of genetically identified individuals in India (joint distribution of the lineage) or a consequence of a local extinction produced by the collision of India with Asia (disjunct distribution of the lineage) [54], and further genetic sequences need to be analyzed to clarify it.

## 5. Conclusions

Our hypotheses, supported by the fossil record [48–50,52], point to the origin of *Aurelia*'s antecessor in the Paleo-Tethys Ocean, from which it spread to the adjacent basins before human intervention. The current biogeography of the genus (Figure 4), the paleogeographic approximation (Figures 5 and 6), and the review of the literature contemporaneous with the opening of the Suez Canal (Table 4) all suggest that *A. solida* is almost certainly native to the Mediterranean Sea. This work aligns with previous suggestions by Lawley et al. [1], Schäfer et al. [58], and Moura et al. [14] to reclassify *A. solida* in the Mediterranean Sea and remove its status as NIS. However, as it is still necessary to conduct genetic analysis on the individuals of Maldives, the type locality of *A. solida*, and to maintain consistency with the previous studies [11,23,58,59], we suggest keeping the current name for the time being instead of resurrecting one of the alternatives listed in Table 4. This work also suggests that *A. coerulea* could also be autochthonous from the Tethys Ocean, with its natural distribution range encompassing Europe and the Mediterranean Sea, potentially negating its classification as NIS.

**Supplementary Materials:** The following supporting information can be downloaded at: https://www.mdpi.com/article/10.3390/d15121181/s1, Figure S1: Non-collapsed maximum likelihood phylogenetic tree; Figure S2: Non-collapsed Bayesian inference phylogenetic tree; Table S1: Genetic distances between individuals of *Aurelia solida* from the Mar Menor coastal lagoon; Table S2: Coordinates where the genetically identified individuals of the genus *Aurelia* have been collected.

**Author Contributions:** Conceptualization, A.F.-A. and A.P.-R.; methodology, A.F.-A.; validation, A.F.-A., C.M. and A.P.-R.; formal analysis, A.F.-A. and A.P.-R.; investigation, A.F.-A.; data curation, A.F.-A.; writing—original draft preparation, A.F.-A.; writing—review and editing, A.F.-A., C.M. and A.P.-R.; visualization, A.F.-A. and A.P.-R.; supervision, C.M. and A.P.-R.; project administration, C.M. and A.P.-R.; funding acquisition, C.M. and A.P.-R. All authors have read and agreed to the published version of the manuscript.

**Funding:** This study has been supported by the project Monitoring the Ecological State of the Mar Menor funded by the General Directorate of the Mar Menor of the Community of the Region of Murcia. A.F.-A. was supported by Fundación Séneca, Región de Murcia (Spain), grant number 21449/FPI/20.

**Data Availability Statement:** The genetic sequences generated in this study have been uploaded to GenBank and the accession numbers are provided in the article. The rest of the data is provided in the main body of the article or as supplementary material.

**Acknowledgments:** This study was conducted as part of the Ph.D. dissertation of A.F.-A. and was supported by 21449/FPI/20, Fundación Seneca, Región de Murcia (Spain). This work has also benefited from data and information from the different projects on "Monitoring and predictive analysis of the ecological state evolution of the Mar Menor lagoon ecosystem and prevention of impacts (2016–2023)" financed by the General Directorate of the Mar Menor of the Autonomous Community of the Region of Murcia. We are grateful to Mar Torralva Forero, from the Zoology Department at the University of Murcia (Spain), who provided us the *Aurelia solida* specimen collected in 1987 for its genetic analysis, and to Andrés García-Reina for his comments on the same. We extend our gratitude to two anonymous reviewers and the editor for their suggestions on the manuscript.

**Conflicts of Interest:** The authors declare no conflict of interest.

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
