# Peer review of "Reconstructing the Biogeographic History of the Genus Aurelia Lamarck, 1816 (Cnidaria, Scyphozoa), and Reassessing the Nonindigenous Status of A. solida and A. coerulea in the Mediterranean Sea"

_diversity, doi:10.3390/d15121181_

Round 1

Reviewer 1 Report

Comments and Suggestions for Authors

The research article is an interesting reconstruction of the biogeographic history of the genus Aurelia which attempts to reassess the non-indigenous status of some Mediterranean species. The introduction is well written and the aims of the research are clearly stated. The investigation tries to explain the origins of the most common species of Aurelia spp. using the available genetic database and an additional genetic analysis performed in the lagoon of Mar Menor (Southern Spain) together with a reconstruction of the paleogeography of the ocean basins where the ancestor of Aurelia species originated.

The authors propose two hypotheses to explain the disjunct distribution of certain species. One hypothesis involves the introduction of Aurelia spp. mainly by anthropogenic activities and the other one focuses on natural processes that occurred through the geological history of the oceans.

Both hypotheses, supported by fossil and historical records, suggest a that Aurelia spp. originated in the Paleo-Tethys from where it migrated the adjacent and interconnected basins.

On the basis of available data of fossil records, literature and genetic analysis the authors discussed whether certain species of Aurelia can be considered autochthonous or allochthonous in the Mediterranean and concluded that Aurelia solida should not be considered an allochthonous species in the Mediterranean.

The materials and methods described in the article are appropriate and clearly explained. The results are extensively described and maybe could be shortened a bit to avoid some repetitions.

The figures 4, 5 and 6 even if they deliver the message of the authors, they could be improved to make them better. What I mean is to consider placing a mask of the continents contours above the areas that indicate the species present. After all the species are marine organisms and thus their presence should be shown only in the marine environment.

The discussion is well argumented and supports the conclusions. I think that the section “Final remarks” should be merged and diluted with the rest of the discussion and that a more concise conclusion paragraph should be written.

Overall, I think it is a good article and I would recommend its publication after a minor revision.

Some little additional specific comments are:

In figure 1 the geographic grid could be removed and only the coordinates at the map margins would be sufficient to identify the area and the addition of a legend on the map would make it easier to understand instead of the text describing the figure.

Line 221: instead of “retrieve” it should be “retrieved”.

In figure 3 you should correct the name “Mjlet lakes” with “Mljet lakes”.

Line 585: instead of “of reclassify” it should be “to reclassify”.

Comments on the Quality of English Language

The research article is an interesting reconstruction of the biogeographic history of the genus Aurelia which attempts to reassess the non-indigenous status of some Mediterranean species. The introduction is well written and the aims of the research are clearly stated. The investigation tries to explain the origins of the most common species of Aurelia spp. using the available genetic database and an additional genetic analysis performed in the lagoon of Mar Menor (Southern Spain) together with a reconstruction of the paleogeography of the ocean basins where the ancestor of Aurelia species originated.

The authors propose two hypotheses to explain the disjunct distribution of certain species. One hypothesis involves the introduction of Aurelia spp. mainly by anthropogenic activities and the other one focuses on natural processes that occurred through the geological history of the oceans.

Both hypotheses, supported by fossil and historical records, suggest a that Aurelia spp. originated in the Paleo-Tethys from where it migrated the adjacent and interconnected basins.

On the basis of available data of fossil records, literature and genetic analysis the authors discussed whether certain species of Aurelia can be considered autochthonous or allochthonous in the Mediterranean and concluded that Aurelia solida should not be considered an allochthonous species in the Mediterranean.

The materials and methods described in the article are appropriate and clearly explained. The results are extensively described and maybe could be shortened a bit to avoid some repetitions.

The figures 4, 5 and 6 even if they deliver the message of the authors, they could be improved to make them better. What I mean is to consider placing a mask of the continents contours above the areas that indicate the species present. After all the species are marine organisms and thus their presence should be shown only in the marine environment.

The discussion is well argumented and supports the conclusions. I think that the section “Final remarks” should be merged and diluted with the rest of the discussion and that a more concise conclusion paragraph should be written.

I think it is a good article and I would recommend its publication after a minor revision.

Some little additional specific comments are:

In figure 1 the geographic grid could be removed and only the coordinates at the map margins would be sufficient to identify the area and the addition of a legend on the map would make it easier to understand instead of the text describing the figure.

Line 221: instead of “retrieve” it should be “retrieved”.

In figure 3 you should correct the name “Mjlet lakes” with “Mljet lakes”.

Line 585: instead of “of reclassify” it should be “to reclassify”.

Reviewer 2 Report

Comments and Suggestions for Authors

Overall a very nice work with some points that could be considered for further improvement:

Line 28: (Lawley et al., 2021) Citations must be kept in accordance with the journal's standards - in the text, citations should be numbered with reference numbers in square brackets.

Line 139: Aurelia sp. into Aurelia sp.

Line 603-730: References must be numbered in the order in which they appear in the text and formatted according to the journal's guidelines.
